# New Insights of SF1 Neurons in Hypothalamic Regulation of Obesity and Diabetes

**DOI:** 10.3390/ijms22126186

**Published:** 2021-06-08

**Authors:** Anna Fosch, Sebastián Zagmutt, Núria Casals, Rosalía Rodríguez-Rodríguez

**Affiliations:** 1Basic Sciences Department, Faculty of Medicine and Health Sciences, Universitat Internacional de Catalunya, 08195 Sant Cugat del Vallès, Spain; afosch@uic.es (A.F.); szagmutt@uic.es (S.Z.); ncasals@uic.es (N.C.); 2Centro de Investigación Biomédica en Red de Fisiopatología de la Obesidad y la Nutrición (CIBEROBN), Instituto de Salud Carlos III, 28029 Madrid, Spain

**Keywords:** SF1 neurons, ventromedial hypothalamus nucleus, obesity, diabetes, energy homeostasis, glucose homeostasis

## Abstract

Despite the substantial role played by the hypothalamus in the regulation of energy balance and glucose homeostasis, the exact mechanisms and neuronal circuits underlying this regulation remain poorly understood. In the last 15 years, investigations using transgenic models, optogenetic, and chemogenetic approaches have revealed that SF1 neurons in the ventromedial hypothalamus are a specific lead in the brain’s ability to sense glucose levels and conduct insulin and leptin signaling in energy expenditure and glucose homeostasis, with minor feeding control. Deletion of hormonal receptors, nutritional sensors, or synaptic receptors in SF1 neurons triggers metabolic alterations mostly appreciated under high-fat feeding, indicating that SF1 neurons are particularly important for metabolic adaptation in the early stages of obesity. Although these studies have provided exciting insight into the implications of hypothalamic SF1 neurons on whole-body energy homeostasis, new questions have arisen from these results. Particularly, the existence of neuronal sub-populations of SF1 neurons and the intricate neurocircuitry linking these neurons with other nuclei and with the periphery. In this review, we address the most relevant studies carried out in SF1 neurons to date, to provide a global view of the central role played by these neurons in the pathogenesis of obesity and diabetes.

## 1. Introduction

Obesity is a multifactorial chronic disease associated with a higher risk of developing cardiovascular diseases, diabetes, cancer, and, more recently, COVID-19 infection. According to the World Health Organization (WHO), in 2016, worldwide obesity had nearly tripled since 1975, with 39% of adults and 18% of children and adolescents overweight or obese [1]. The main metabolic comorbidity of obesity is type-2 diabetes that occurs when body tissues become resistant to insulin and is estimated to be the seventh leading cause of death [2]. Therefore, understanding the molecular and physiological mechanisms underlying the control of feeding behavior, energy balance and glucose homeostasis is crucial for the prevention and treatment of obesity and diabetes.

The regulation of peripheral metabolism and glucose homeostasis not only resides in the tissue’s autonomous responses to nutrients and hormonal signals but also in some brain regions, particularly the hypothalamus. The hypothalamus integrates multiple metabolic inputs from peripheral organs with afferent stimuli coming from other brain regions and coordinates a diversity of efferent responses to control food intake, fat metabolism, hormone secretion, body temperature, locomotion, and behavior in order to maintain energy balance and blood glucose levels. Within the hypothalamus, the ventromedial nucleus (VMH) located above the arcuate nucleus (ARC) and the median eminence, was identified in the mid-1900s as the satiety center because its injury produced hyperphagia, insulin resistance, and body weight gain [3,4]. At that point, VMH was demonstrated to play a key role in the control of energy expenditure and glucose homeostasis [3,5]. Since then, intensive research has been done on VMH and it is currently known that this hypothalamic area encompasses a heterogeneous set of neurons, which are differentiated by the genes they are expressing (Figure 1). Many of the genes highly expressed in the VMH have been identified and their functions have been explored (Figure 1) [6,7,8].

The majority of VMH cells, especially in the dorsomedial and central regions of VMH (VMHdm and VMHc, respectively) [9], express the nuclear receptor steroidogenic factor 1 (SF1) (Figure 1), also known as NR5A1, which is essential for VMH development and function [10,11,12] and is considered a key marker of that nucleus since SF1 expression is specific and selective to VMH in the brain. Outside the brain, SF1 can be found in the adrenal gland and gonads with differential roles during development [13]. For many years, SF1 was classified as an orphan nuclear receptor, but between 2005 and 2007 several pieces of evidence demonstrated that phospholipids can be ligands of SF1, such as phosphatidic acid being an activator and sphingosine an inhibitor of SF1 transcriptional activity [13]. Interestingly, SF1 can suffer different post-translational modifications, which regulate its stability and transcriptional activity [14], but also control the expression of numerous downstream target genes, including CB1, BDNF, and Crhr2 [13]. Considering this, in order to explore the importance of SF1 neurons, transgenic mice lacking this nuclear receptor were studied by different researchers. Mice lacking SF1 were not viable due to a failure in the proper development of adrenal glands and gonads [11,13,15]. However, when rescued from lethality by adrenal transplantation from WT littermates and corticosteroid injections, mutant mice displayed robust weight gain resulting from both hyperphagia and reduced energy expenditure [16]. Interestingly, a similar phenotype was observed in humans with mutations of the SF1 gene, who often show mild to severe obesity [17,18]. To avoid the confounding metabolic effects of glucocorticoid injections and adrenal transplantations to global SF1 KO mice, Elmquist’s group generated an alternative mouse model in which SF1 was specifically deleted in the VMH after completion of the nucleus development using CamKII-Cre [19,20]. These postnatal VMH-specific SF1 KO mice showed increased weight gain and impaired thermogenesis in response to a high-fat diet (HFD), being the first demonstration that the transcription factor SF1 is postnatally required in the VMH for normal energy homeostasis, especially under the HFD condition.

In an attempt to clarify the contribution of this specific population of neurons to hypothalamic regulation of obesity and diabetes, in the last 15 years, several transgenic models have been developed by deleting specific targets in SF1 neurons related to energy balance and glucose homeostasis. In 2013, a profound review article was published by Choi and colleagues [8] summarizing the last updates of SF1 neurons in energy homeostasis. Since then, new neuronal-based approaches (i.e., optogenetic and chemogenetic technology) and the generation of new transgenic mice in key target proteins have provided exciting insight into the implication of SF1 neurons on whole-body energy balance, particularly thermogenesis and glucose homeostasis, that are compiled in the present review.

Despite these studies contributing to understanding the mechanisms by which SF1 neurons regulate energy homeostasis, new questions arose from these results. An important issue to clarify is the heterogeneity of SF1 neurons in terms of glucose, leptin, and insulin sensing or non-sensing neurons, indicating the operation complexity of SF1 neurons and the intricate mechanisms of afferent and efferent neurocircuitry involved in the control of energy balance. In this review, we address the most relevant studies carried out on SF1 neurons to date, from transgenic mice to neurocircuitry studies, in order to discuss the most relevant findings of these investigations and provide an exhaustive overview of the role played by SF1 neurons in the hypothalamic regulation of energy expenditure and glucose homeostasis, and the potential mechanisms involved.

## 2. Unraveling the Functions of SF1 Neurons in Energy Balance by Optogenetic and Chemogenetic Approaches

In order to selectively manipulate the SF1 neuronal activity in a physiological context, optogenetic and chemogenetic approaches have emerged [21,22]. These tools use channels that are activated by light and engineered G-protein coupled receptors controlled by exogenous molecules, respectively [21,22]. The incorporation of these approaches into animal models has greatly advanced our understanding of the SF1 role and neuronal circuits.

Mice engineered to activate SF1 neurons by optogenetics were designed through the injection of adeno-associated virus (AAV) particles expressing a Cre-dependent channelrhodopsin (ChRs) into the VMH of SF1-Cre mice to produce SF1-ChRs animals [23,24]. The advantage of this technology is the light-controllable activation of SF1 neurons in a spatiotemporal manner.

One of the first results obtained from this model revealed that SF1 neurons are closely associated with defensive reactions. David J. Anderson and colleagues in 2015, demonstrated that the optogenetic stimulation of SF1 neurons applying a frequency of 20 Hz induced freezing or activity burst, while no response on feeding behavior or energy balance was reported [25]. However, the electrophysiological record of the firing pattern observed in these studies was generated by applying a high-frequency burst of spiking, while the physiological spiking of the neurons in steady-state ranged between 3.5–6.3 Hz [26]. Considering this information, two years later, other researchers showed that SF1 neurons exert a differential effect depending on the frequency of activation. They confirmed that high-frequency activation (>20 Hz) evokes a profound defensive response which includes freezing and escape attempts, but low-frequency activation (2 Hz) suppresses feeding after fasting and reduces the time that mice spend near to the food [24]. These novel results suggest that SF1 neurons dynamically modulate feeding and anxiety-related behaviors by changing the firing pattern and also indicate that this subset of hypothalamic neurons is involved in the fight or flight response.

The chemogenetic modulation of SF1 neurons under potentially more physiological firing patterns has been performed using hM3Dq and hM4Di designer receptors exclusively activated by designer drugs (DREADDs). The expression of DREADDs specifically in SF1 neurons allows to excite or inhibit them in response to clozapine (CNO) or JHU37160 dihydrochloride (J60) which is an in vivo DREADD agonist with high affinity and potency for hM3Dq and hM4Di receptors [27,28]. In agreement with the feeding behavior-related results obtained from optogenetic approaches, the administration of CNO in fasted SF1-hM3Dq mice revealed a reduction in food consumption. On the contrary, inhibition of SF1 neurons in SF1-hM4Di mice increased cumulative food intake in ad libitum-fed mice. Authors manifested that none of the dramatic defensive behaviors seen with high-frequency optogenetic activation of SF1 neurons was observed [24]. At peripheral levels, they also demonstrated that the chronic chemogenetic modulation of SF1 neurons modified the body fat mass since the continuous administration of CNO (3 weeks) in SF1-hM3Dq and SF1-hM4Di animals reduced and increased the fat mass content, respectively [24]. These changes in fat mass could be explained by the SF1 modulation of fat oxidation. Very recently, it was established that SF1-hM3Dq mice increased energy expenditure and fat oxidation independent of the locomotion activity within 2-h post-activation [23]. Although this study did not evaluate the energy expenditure profile in SF1-hM4Di mice, it was expected that the inactivation of SF1 neurons reduced energy expenditure. This hypothesis is supported by mice expressing tetanus toxin (TT) in SF1 neurons. Since TT prevents neurotransmitter release, SF^TT^ mice displayed reduced energy expenditure and increased body weight [29].

Besides their implication in energy balance, the use of optogenetics has highlighted the role of SF1 neurons in the hypothalamic control of systemic glucose levels. For a long time, it was known that VMH triggered the counterregulatory response (CRR) induced by hypoglycemia [30,31] but it was not clear the contribution of SF1 neurons to this feedback response. In order to elucidate whether SF1 neurons are linked to this effect, an elegant experiment using optogenetic and chemogenetic tools was done by Gregory J Morton and colleagues, showing that selective inhibition of SF1 neurons blocked recovery from insulin-induced hypoglycemia. Conversely, activation of SF1 neurons caused diabetes-range hyperglycemia [32]. This evidence is concordant with those obtained from transgenic models such as mice lacking vesicular glutamate transporter 2 (VGLUT2) specifically in SF1 neurons since this genetic disruption attenuated recovery from insulin-induced hypoglycemia [33].

Taken all together, these genetic approaches have revealed the specific involvement of SF1 neurons in many aspects of metabolic regulation due to their direct or indirect role in the maintenance of the energy balance and glucose levels, confirming the classification of VMH as a primary satiety center [34,35].

## 3. Manipulation of Key Targets in SF1 Neurons: Lessons from Transgenic Mice

A particularly powerful strategy developed for the exploration of SF1 neurons in obesity and diabetes has been the design of the SF1 Cre mice. Several groups have generated different SF1 Cre transgenic lines in which the expression of Cre recombinase is derived by *Sf1* regulatory elements [20,36]. These lines allow for ablating general factors or targets known to be associated with energy homeostasis by crossing them with *floxed* strains. In the following sub-sections, we discuss different studies of SF1-CRE transgenic mice organized by the type of molecular target under investigation (Table 1) as well as the sex-specific effect of SF1 neurons in energy balance.

### 3.1. Hormone Receptors and Related Signaling Pathways

Due to the importance of the anorectic hormone leptin in the central control of energy homeostasis, the physiological effects after deletion of leptin receptor (LEPR) in SF1 neurons have been thoroughly investigated. The first and most representative study was performed by Bradford Lowell and colleagues, in which genetic deletion of LEPR selectively from hypothalamic SF1 neurons triggered an increase in body weight gain without changes in food intake, leaving these mice unable to adapt to HFD or to activate energy expenditure [20]. To reinforce the key role of the leptin pathway in SF1 neurons, selective inactivation of *Socs3*, a negative mediator of central leptin—pSTAT3 signaling, in SF1 neurons (Figure 2) was developed. Conversely to the deletion of LEPR, mice lacking *Socs3* showed improved weight-reducing effects of leptin, with a decrease in food intake and an enhanced energy expenditure under chow diet or HFD condition [37]. The importance of leptin signaling in energy balance through SF1 neurons was also reinforced by the specific deletion of the G protein α-subunit Gsα [38]. Gsα couple receptors for hormones, neurotransmitters, and other factors to activate adenyl-cyclase leading to cAMP generation, which is a negative regulator of leptin action (Figure 2). Then, the lack of Gsα in SF1 neurons increases leptin sensitivity [38].

The protein-tyrosine phosphatase 1B (PTP1B) is another negative regulator of leptin signaling in SF1 neurons (Figure 2). Its action is mediated through selective dephosphorylation of the two signaling molecules JAK2 and STAT3. In vivo studies have demonstrated that whole-brain deletion of PTP1B resulted in leanness, hypersensitivity to leptin, and resistance to HFD-induced obesity, a phenotype partly associated with increased hypothalamic activation of STAT3 [39]. Surprisingly, its specific deletion in SF1 neurons resulted in increased adiposity in female mice exposed to HFD due to low energy expenditure, whereas leptin sensitivity was enhanced, and food intake was attenuated, findings that were likely explained by increased STAT3 activation [40]. Mice lacking PTP1B in SF1 neurons also had improved leptin and insulin signaling in VMH, suggesting that increased insulin responsiveness in SF1 neurons could overcome leptin hypersensitivity and promote adiposity [39,40].

A more recent study tried to rescue native LEPR in SF1 neurons in *LepR*-deficient mice. They concluded that LEPR signaling in the VMH is not sufficient to protect against obesity in this null mouse [41]. This finding could explain that this neuronal population expressing LEPR works in conjunction with other types of neurons expressing the same receptor, and SF1 neurons by themselves cannot compensate for all receptor deficiency. Summing up, leptin signaling in SF1 neurons plays a key role in energy homeostasis regulation and mediates the proper physiological adaptation to HFD to avoid or delay the onset of obesity.

According to glucose metabolism, leptin has been long related to glucose homeostasis improving insulin sensitivity, since intra-VMH injection of leptin increases glucose uptake in peripheral tissues [42] and normalizes hyperglycemia [43]. Very recently it has been demonstrated that central leptin infusion in mice with SF1 neuron-specific LEPR deficiency corrected diabetic hyperglycemia [43]. Despite this result, re-expressing the receptor in SF1 neurons of null mice showed that SF1 neurons were not sufficient to mediate the antidiabetic action of leptin [44]. The essential action of leptin in SF1 to correct diabetic hyperglycemia was clarified by further investigations. Particularly, in the specific knock—out of *Socs3* in SF1 neurons, where leptin signaling is over-activated, Ren Zhang and colleagues observed improved glucose homeostasis, showing protection against hyperglycemia and hyperinsulinemia caused by HFD feeding [37]. These studies demonstrate that leptin in VMH neurons improves glucose and insulin metabolism, although this area is not essential as there are redundant neuronal circuits regulating it. Optogenetic activation of SF1 neurons has the same output as leptin increasing glucose uptake but it does not normalize blood glucose levels, which leads to an understanding of two different subsets of SF1 neurons, one subset increasing insulin sensitivity and the second one increasing blood glucose levels.

Another key hormone implicated in the balancing of energy metabolism is insulin. It is known that insulin acutely suppresses food intake and decreases fat mass in both rodents and humans [45,46] (Figure 2). Mice lacking insulin receptors (IR) in SF1 neurons did not show any differences in body weight when fed a chow diet but under HFD conditions, mutant mice were protected against obesity and showed an enhanced leptin sensitivity and glucose homeostasis [26]. Interestingly, exposure to HFD led to the overactivation of insulin in the VMH, leading to a reduction in SF1 neurons firing frequency, in comparison to the insulin resistance induced in ARC neurons. The differential dysregulation of insulin action in these hypothalamic nuclei under HFD could indicate cooperation between these responses to drive obesity [26]. These findings also suggest that hyperinsulinemia evoked by HFD feeding controls SF1 neuronal activity, which leads to changes in the synaptic inputs to other neuronal populations such as POMC, resulting in obesity and diabetes. However, the specific contribution of insulin signaling in SF1 neurons and its relationship to peripheral insulin resistance and glucose levels needs further investigation.

Leptin depolarizes or hyperpolarizes SF1 neurons, depending on the subpopulation, while insulin only hyperpolarizes them, as both actions are in a downstream PI3K-dependent manner [20,47,48,49] (Figure 2). PI3K is formed by two different subunits, p85 and p110, and specifically, the subunit p110β is necessary for depolarizing and hyperpolarizing SF1 neurons, while p110α is only needed in the hyperpolarization process [47]. Mice lacking p110α in SF1 neurons had reduced energy expenditure in response to hypercaloric feeding and, therefore, displayed an obesogenic phenotype. Moreover, this p110α subunit was not required to regulate glucose metabolism in SF1 neurons [49]. Mice lacking p110β in the same neuronal population had also decreased energy expenditure (reduced thermogenesis) leading to increased susceptibility to obesity, whereas, in contrast to the p110α subunit, p110β involved changes in peripheral insulin sensitivity [50]. In line with this evidence, deletion of FOXO1, a downstream transcription factor of insulin-PI3K (Figure 2), in SF1 neurons resulted in a lean phenotype with high energy expenditure, even in fasting, and these null mice presented an enhanced insulin sensitivity and glucose tolerance, in concordance with genetic deletion of IR in SF1neurons [51]. Although leptin and insulin can inhibit SF1 neurons using the same molecular cascade, they are anatomically segregated within the VMH (neurons expressing LEPR receptor are located in the VMHdm when depolarizing and scattered throughout the nucleus when hyperpolarizing, whereas those expressing IR are in the VMHc close to the ventricle) [47], which would explain the different effects observed when deleting their receptors.

Other hormones studied in SF1 neurons are estrogens. Female mice lacking the estrogenic receptor α (ERα) in SF1 neurons were obese due to a reduced energy expenditure [52]. Ablation of the ERα led to abdominal obesity with adipocyte hypertrophy in females, but not in male mice [53]. Despite the fact that most of the studies on SF1 neurons until now were performed only in male mice, these last results described, and others discussed later [40,54], reinforce the notion that SF1 neurons may have a sex-specific effect on energy balance and glucose metabolism.

Growth hormone (GH) also plays a role in glucose metabolism via SF1 neurons. GH is secreted in a metabolic stress situation such as hypoglycemia. Deletion of its receptor in SF1 neurons resulted in an impaired capacity for recovery from hypoglycemia [55]. These mice showed an altered CRR due to changes in the neurocircuit that regulates the parasympathetic nervous system. This result supports the importance of SF1 neurons in the proper functionality of glucose homeostasis.

Altogether, these findings identify SF1 neurons (the predominant VMH population) as a key player in the regulation of energy expenditure and glucose homeostasis, being particularly important in the adaptive response to HFD feeding. Most of the mutant mice with deletion of several hormone receptors and associated proteins in SF1 neurons have no changes or mild metabolic alterations under chow diet, but they show substantial metabolic variations under HFD exposure. The action of hormones and related proteins in SF1 neurons is also involved in the CRR to hypoglycemia to maintain glucose balance between the brain and the periphery. Future studies are needed to describe the specific molecular mechanisms and subsets of SF1 neurons underlying the effects of hormones in glucose homeostasis and energy expenditure.

**Table 1 ijms-22-06186-t001:** Genetic models developed to study SF1 neurons in energy balance.

Type of Target	Target	Mice Model Name	Sex	Challenge	BW	FI	EE	Adiposity	Glycemia	Glucose Tolerance	Insulin Sensitivity	Leptin Sensitivity	SNS Activity	Ref.
Hormone receptors and related signaling pathways	LEPR	Sf1-Cre, *Lepr*^flox/flox^	M	SD	↑	n.s.	n.s.	↑	n.s.	-	-	-	-	[20]
				HFD	↑	↑	↓	↑	n.s.	-	-	-	-	
	SOCS3	Sf1-Cre, *Socs3*^flox/flox^	M	SD	n.s.	↓	↓	-	↓	↑	↑	↑	-	[37]
				HF-HS	n.s.	↓	↓	-	↓	↑	↑	↑	-	
				Leptin ^(a)^	↓	↓	-	-	-	-	-	↑	-	
	G_s_α	VMHGsKO	M ^(b)^	SD	n.s.	n.s.	n.s.	-	n.s.	n.s.	n.s.	n.s.	-	[38]
				HFD	n.s.	n.s.	n.s.	-	↓	↑	↑	↑	-	
	PTP1B	Sf1-*Ptpn1*^−/−^	F	HFD	↑	↓	↓	↑	-	-	↑	↑	↓	[39]
			M	HFD	n.s.	-	-	n.s.	-	-	-	-	-	
	IR	SF-1^ΔIR^		SD	n.s.	n.s.	n.s.	n.s.	-	-	-	-	-	[26]
				HFD	↓	↓	n.s.	↓	n.s.	↑	↑	↑	-	
	p110α	p110α^lox/lox^/SF1-Cre	M	SD	n.s.	n.s.	n.s.	n.s.	n.s.	-	n.s.	↓	-	[49]
				HFD	↑	n.s.	↓	↑	-	-	-	-	-	
	p110β	p110β KO^sf1^	M	SD	n.s.	n.s.	↓ BAT th.^(c)^	n.s.	n.s.	↓	↓	-	-	[50]
				HFD	↑	n.s.	↓	↑	↑	-	-	-	-	
	FOXO1	*Foxo1* KO^Sf1^	M	SD	↓	n.s.	↑	↓	-	-	-	-	-	[51]
			F	SD	↓	n.s.	↑	↓	-	-	-	-	-	
			M	HFD	↓	n.s.	↑	↓	↓	↑	↑	↑	-	
	ERα	ERα^lox/lox^/SF1-Cre	F	SD	↑	n.s.	↓	↑	n.s.	↓	-	-	-	[52]
			F	HFD	↑	n.s.	↓	↑	-	-	-	-	↓	
Nutrient sensors	AMPK	SF1-Cre AMPKα1^flox/flox^	M	SD	↓	n.s.	↑	↓	-	-	-	-	↑ ^(d)^	[56]
			M	HFD	↓	n.s.	↑	↓	↓	↑	n.s.	-	-	
	SIRT1	Sf1-Cre; *Sirt1*^loxP/loxP^	M/F	SD	n.s.	n.s.	n.s.	n.s.	n.s.	-	-	-	-	[57]
				HFD	↑	n.s.	↓	↑	↑	↓	↓	↓	-	
Glutamatergic neurotransmission and synaptic receptors	VGLUT2	Sf1-Cre; *Vglut2*^flox/flox^	M/F	SD	n.s.	-	-	-	↓	-	-	-	-	[33]
			M/F	HFD	↑	↑	n.s.	↑	-	-	-	-	-	
	mGluR5	mGluR5^2L/2L:SF1-Cre^	F	SD	n.s.	n.s.	n.s.	-	n.s.	↓	↓	-	↓	[54]
			M	SD	n.s.	n.s.	n.s.	-	n.s.	n.s.	n.s.	n.s.	n.s.	
			M/F	HFD	n.s.	n.s.	n.s.	-	-	-	-	-	-	
	α2δ-1	α2δ-1^2L/2L:SF1-Cre^	M	SD	n.s.	n.s.	n.s.	n.s.	n.s.	↓	↓	-	↓	[58,59]
			M	HFD	n.s.	n.s.	n.s.	n.s.	n.s.	n.s.	n.s.	-	-	
			F	SD	n.s.	n.s.	n.s.	n.s.		↓	↓	-	-	
			F	HFD	↑	n.s.	n.s.	n.s.	n.s.	↓	↓	-	↑	
	CB1	SF1-CB_1_-KO	M	SD	n.s.	n.s.	↑ BAT th.	↓	n.s.	↑	↑	↑	↑	[60]
			M	HFD	↑	↑	n.s.	↑	-	↓	n.s.	↓	↓	
Modulators of autophagy, mitochondrial and primary cilia function	*Atfg7*	Sf1-Cre; *Atg7*^loxP/loxP^	M	Fasting	n.s.	↓	↓	n.s.	n.s.	n.s.	n.s.	↓	-	[61]
	UCP2	*Ucp2*KOKI^Sf1^	M	Chow diet	n.s.	n.s.	n.s.	n.s.	n.s.	↑	↑	-	-	[62]
	IFT88	IFT88-KO^SF−1^	M/F	Chow diet	↑	n.s.	↓	↑	↑	↓	↓	↓	↓	[63]
			M/F	HFD	↑	↑	↓	↑	↑	-	-	-	-	

n.s.: No significant changes appreciated; -: not studied/unknown; M: male; F: female. SD: standard diet; HFD: high fat diet; HF-HS: high fat-high sucrose diet; BAT th.: brown fat thermogenesis. ^(a)^ Subcutaneously implanted osmotic minipumps, infusing for 14 days at 0,5 µg/h. ^(b)^ The study was performed in both male and female, but the metabolic alterations were only appreciated in male mice. ^(c)^ No changes in EE but significant decrease in BAT thermogenesis. ^(d)^ Increased sympathetic activity in brown fat.

### 3.2. Nutrient Sensors: AMPK and SIRT1

The regulation of energy homeostasis is importantly regulated by proteins acting as nutrient and energy sensors, such as AMPK and SIRT1, which restore energy balance during metabolic challenges both at the cellular and physiological levels (Figure 3). A link between both AMPK and SIRT1 in hypothalamic SF1 neurons with central control of obesity and diabetes has been established.

AMPK is a highly conserved master regulator of metabolism, which is activated under low energy conditions, increasing energy production and reducing energy waste. Current evidence demonstrates the critical role of hypothalamic AMPK in the regulation of food intake and energy expenditure, as well as glucose and lipid homeostasis at the whole-body level [56,64,65,66,67]. Interestingly, the regulation of these effects depends on the anatomical location of AMPK in the hypothalamus: AMPK effects in feeding control and glucose homeostasis mostly rely on ARC neurons [64,66,67], whereas its effects in energy expenditure arise from the VMH [56,65,68], particularly in SF1 neurons (Figure 3).

The identification of SF1 neurons as the main neuronal AMPK-expressing population in VMH to be targeted in obesity was demonstrated by several studies of Miguel Lopez lab. They generated an SF1 specific AMPKα_1_ null mouse by Cre-Lox recombination. Ablation of AMPKα_1_ in these neurons led to feeding-independent weight loss associated with an increase in energy expenditure [56]. In accordance with this, SF1-Cre AMPKα_1_^flox/flox^ mice displayed BAT activation, which was confirmed by increased BAT temperature and UCP1 protein expression, elevated sympathetic activity, and higher ^18^F-FDG uptake in brown fat [56]. SF1-Cre AMPKα_1_^flox/flox^ mice fed with a long-term HFD had a feeding-independent decrease in body weight and adiposity, associated with increased energy expenditure and VO_2_, higher BAT thermogenesis activation, and browning of subcutaneous WAT. These findings point to AMPKα_1_, but not AMPKα_2_, in SF1 neurons as the main catalytic AMPK subunit in the VMH that regulate thermogenic control, particularly under HFD exposure.

Other studies demonstrating that the stimulatory effect of thyroid hormones in BAT thermogenesis [65], WAT browning, and lipid metabolism are mediated by AMPKα_1_ attenuation in SF1 neurons [68] strongly support the notion that specific targeting of the discrete neuronal population in the VMH impacts obesity in a feeding-independent, but the thermogenic-dependent manner [56]. These studies also suggest that simultaneous targeting of AMPK in SF1 neurons of the VMH and neurons in the ARC would allow for controlling both food intake and energy expenditure by inhibiting a common protein.

Although the protection of AMPKα_1_ deletion in SF1 neurons against HFD-induced obesity was mainly associated with increased energy expenditure, slight alterations in glucose balance (decreased glycemia and improved glucose tolerance without changes in insulin sensitivity and insulin levels) found in mutant mice fed with an HFD, could be involved [56]. The moderate role of AMPK in glucose sensing by SF1 neurons in the VMH was also reported in a recent study in which AMPK activity suppression led to selective depletion of SF1 glucose inhibitory (GI) neurons, and activated CRR without affecting the presence of glucose excited (GE) neurons [69]. The authors suggested that the primary role of AMPK in SF1 GI neurons is to control the expression of *Txn2* (encoding a mitochondrial redox enzyme), providing protection against ROS produced during hypoglycemia. Conversely, under hyperglycemia, a reduction in ROS production is mediated by UCP2 in GE SF1 neurons [62]. These findings support the idea of two different subsets of SF1 neurons: the one regulating hyperglycemia enhancing insulin sensitivity and the second one regulating hypoglycemia by activating CRR.

In addition to AMPK, the metabolic sensor of nicotinamide adenine dinucleotide-dependent deacetylase SIRT1, which is highly expressed in the VMH, has shown to contribute to both control of energy balance and clock function from SF1 neurons [70,71,72]. SIRT1 deacetylates several proteins in central and peripheral tissues to provide adaptation against redox/nutrient challenges. SIRT1 drives lipid mobilization from adipose tissue, lipid oxidation in skeletal muscle and liver, and increased glucose production, all of these responses protect against diminished energy availability [70]. In the hypothalamus, SIRT1 has been revealed as an attractive target against obesity and type 2 diabetes in both POMC [73] and SF1 [57] neurons. SIRT1 activation in these hypothalamic neurons triggers a reduction in body adiposity and insulin resistance in obesogenic environments, while the deleterious effects, such as increased glucose and lipid output from the liver, would be avoided.

As observed with AMPK, the anatomical location of SIRT1 in the hypothalamus is relevant for the protection against diet-induced obesity: although SIRT in POMC neurons is required for normal BAT-like remodeling in the perigonadal fat depot [73], SIRT1 in SF1 neurons controls insulin sensitivity in the skeletal muscle [57] (Figure 3). Mice lacking SIRT1 in SF1 neurons showed hypersensitivity to hypercaloric diets, gaining more bodyweight and lower energy expenditure compared to control mice after long-term HFD, whereas food intake remained unchanged [57].

In addition, to protect against dietary obesity, SIRT1 in SF1 neurons is relevant for homeostatic responses that can protect against dietary diabetes, since mice overexpressing SIRT1 exclusively in SF1 neurons were protected from developing diet-induced insulin resistance in skeletal muscle and hyperglycemia, whereas mice lacking SIRT1 in the same neurons were more prone to develop insulin resistance in skeletal muscle after HFD [57]. It is important to note that these studies indicate that SIRT1 in SF1 neurons does not regulate food intake in either chow or HF feeding and that the regulatory effects on energy expenditure and glucose homeostasis are predominantly observed in a hypercaloric diet but not in a normal diet context. The latter evidence is intriguing since SIRT1 is known to be activated in the low-energy state. According to Coppari [70], the possible explanations are i) HFD increases SIRT1 activity in SF1 neurons, and ii) in chow-fed mice, compensatory mechanisms (i.e., enhancement of insulin sensitivity in the liver to provide normal glycemia) counterbalance the defects related to SIRT1 deficiency in these hypothalamic neurons. These hypotheses could explain that the metabolic imbalance is mainly observed in transgenic mice exposed to an HFD when the compensatory mechanisms are unable to prevent the whole-body effects produced to the lack of SIRT1 in hypothalamic neurons (i.e., insulin sensitivity is reduced in the skeletal muscle but is not increased in the liver).

In addition to these investigations, SIRT1 also contributes to the synchronization of the circadian clock in the brain, by nutritional inputs, which is crucial to maintaining cellular and body homeostasis [71]. Mice with targeted ablation of SIRT1 in SF1 neurons of the VMH showed a reduction in the period length in the light-entrainable activity under constant darkness and scheduled feeding, but not in ad libitum feeding [71]. Furthermore, these effects were more pronounced in KO mice under scheduled high-caloric diet feeding. This investigation suggests that SIRT1 in SF1 neurons contributes to the synchronization and/or adaptation of the endogenous clock to feeding cues (Figure 3), a function that would partially explain the metabolic alterations displayed by mice deficient in SIRT1 in SF1 neurons previously reported [57]. Accordingly, deletion of BMAL1 in SF1 neurons of the VMH in mice also drove a substantial alteration of the circadian clock in the brain, with an impact in BAT activation, supporting the importance of SF1 neurons in the regulation of clock function in the brain and the whole-body energy homeostasis [72]. The implication of SIRT1 in SF1 neurons in healthy aging or lifespan via circadian clock modulation would be also interesting to explore and would provide insight into the hypothalamic mechanisms mediated by SF1 neurons regulating energy balance.

Overall, specific targeting of metabolic sensors such as AMPK and SIRT1 in SF1 neurons of the VMH would be enough to modulate body weight gain and energy expenditure, in a feeding-independent manner, but associated with synchronization of the central clock with the periphery. This is a challenging approach but with significant translational relevance in obesity management.

### 3.3. Glutamatergic Neurotransmission and Synaptic Receptors

The expression and activity of synaptic receptors and glutamatergic neurotransmission in SF1 neurons of the VMH have been identified to contribute to energy balance dysregulation in obesity and diabetes.

Since neurons in the VMH are largely glutamatergic, the importance of glutamate neurotransmission versus other neurotransmitters and neuropeptides in mediating the functions of these nuclei seem to be higher, not only in energy homeostasis but also in sexual and defensive behaviors [8,33]. The first study reporting this evidence in SF1 neurons was provided by Bradford Lowell and colleagues [33]. They generated a mouse lacking the VGLUT2—a transporter mediating uptake of L-glutamate into synaptic vesicles—specifically in SF1 neurons, driving to a disruption in transsynaptic communication by glutamate in this specific population of neurons. The phenotype study was particularly focused on glucose and insulin homeostasis: these mice displayed hypoglycemia in the fasted state secondary to impaired fasting-induced glucagon increased and impaired gluconeogenesis induction in the liver [33] (Figure 4). VGLUT2 deficient mice in SF1 neurons also had defective CRR to insulin-induced hypoglycemia and central 2-deoxyglucose, as this greater degree of hypoglycemia again linked to an impaired glucagon response. It is important to mention that SF1-Cre;Vglut2^flox/flox^ mice did not become obese when exposed to a chow diet and had mild obesity when fed with an HFD [33]. A partial contribution of GABA release, from a small number of SF1 neurons, in body weight regulation in these mice could explain this mild obesogenic phenotype [33]. Importantly, this study was the first demonstration of the importance of glutamate release in VMH neurocircuitry to prevent hypoglycemia.

The relevant role of glutamatergic neurotransmission in SF1 neurons to regulate energy homeostasis was also recently established by the development of mice lacking the metabotropic glutamate receptor mGluR5 in these neurons [54]. The most remarkable finding of this study was that mGluR5 depletion in SF1 neurons did not affect energy balance, but it significantly impaired glucose balance control (Figure 4), as this dysfunction was only observed in female but not in male mice. The sex-specific impairment of insulin sensitivity and glucose tolerance was associated with a reduction in intrinsic excitability and the firing rate of SF1 neurons. This suggests a functional indirect interaction of mGluR5 with estrogen receptors to control the effects of estradiol on SF1 neurons activity and glycemic control, which switch from facilitatory to detrimental in the absence of the glutamate receptor in these neurons [54]. These results are in line with the previously described study reporting that ERα depletion in SF1 neurons drives metabolic disturbances only in females [52]. However, the lack of impairment in insulin-induced hypoglycemia in mice deficient in mGluR5 is in contrast to the previous investigation in which inhibition of glutamate release by SF1 neurons impaired the CRR to hypoglycemia [33].

Other receptors involved in neuronal activity have shown their contribution to the regulation of obesity and diabetes by SF1 neurons in the VMH. Among them, the calcium channel subunit α2δ-1 has a non-canonical role in SF1 neurons to control glucose and lipid homeostasis (Figure 4). This neuronal receptor facilitates cell surface trafficking of calcium channels mediating calcium current and neurotransmitters release [74] and also acts as a receptor for thrombospondins, promoting excitatory synapse assembly [75]. The authors show that α2δ-1 is essential for the excitability and firing activity of SF1 neurons. Deletion of this receptor in SF1 neurons in mice led to neuronal hypoactivity, sympathetic tone reduction to WAT and skeletal muscle, glucose intolerance, and insulin resistance, and lipolysis alteration in WAT [58], as these metabolic alterations mainly observed in female mice [59]. Despite these findings, mice did not display alterations in food intake and body weight in response to HFD [58]. The reduction in SF1 neuronal activity observed in this transgenic mouse did not compromise CRR to hypoglycemia, in contrast to previous studies in which chemogenic or optogenetic silencing of these neurons impaired recovery from hypoglycemia [32,76]. These conflicting results could be related to the heterogeneity of SF1 neurons, with functional diversity in the VMH (e.g., leptin-activated and leptin-inhibited SF1 neurons) [47].

The endocannabinoid receptor CB1 located in SF1 neurons of the VMH has been also involved in glutamatergic output regulating feeding and in the control of metabolic flexibility in response to dietary changes. Hypothalamic slice preparations exposed to a CB1 receptor-specific agonist, displayed inhibition of the electrical activity of SF1 neurons [77]. In line with this, endocannabinoids release in postsynaptic neurons, led to retrograde activation of CB1 receptors in pre-synaptic glutamatergic SF1 neurons of the VMH, blocking calcium entry with subsequent inhibition of glutamatergic output and attenuation of POMC neurons activation, promoting feeding in diet-induced obesity [78] (Figure 4). The physiological relevance of CB1 receptors in SF1 neurons was also reported by Daniela Cota and colleagues, showing that mice lacking CB1 receptor specifically in these neurons showed a diet-dependent bidirectional metabolic phenotype: under a chow diet, deletion of CB1 decreased adiposity by increasing sympathetic activity and WAT lipolysis, whether, conversely, under HFD, lack of this receptor in SF1 neurons blunted peripheral use of lipids and produced leptin resistance, hyperphagia, body weight gain and glucose intolerance [60]. Therefore, CB1 synaptic receptor in SF1 neurons seems to act as a molecular switch for correct metabolic flexibility in the hypothalamus, protecting from the development of HFD-induced leptin resistance.

Altogether, although these studies provide substantial evidence of the role of glutamatergic neurotransmission and neuronal receptors in SF1 neurons to control glucose homeostasis and metabolic adaptation to dietary challenges, they raise important questions that still need to be clarified such as (i) the identification of the exact afferent and efferent components of the neurocircuitry linking SF1 neurons with other brain areas and the periphery (e.g., for the regulation of glucagon release in the pancreas), (ii) the evaluation of sex-dependent function, (iii) the exploration of SF1 neuronal subpopulations in response to different stimuli (e.g., important for the compensatory effects on feeding or the CRR to hypoglycemia).

### 3.4. Modulators of Autophagy, Mitochondrial and Primary Cilia Function

Autophagy plays a critical role in several physiological processes such as metabolic regulations. Deletion of essential autophagy genes, such as the autophagy-related gene (Atg) 7 in peripheral tissues drives significant changes in body weight and glucose balance [79,80]. More recent investigations have demonstrated that autophagy is important in the hypothalamic regulation of energy homeostasis and feeding behavior, with these effects affected differently depending on the neuronal types engaged. In the ARC, *Atg7* deletion in POMC neurons implied an obesogenic phenotype, especially when feeding an HFD [81,82], whereas loss of this gene in AgRP neurons promoted leanness [83]. A more recent study reported the metabolic importance of autophagy in the VMH, by deleting *Atg7* in SF1 VMH neurons of mice [61]. The authors found that in fed conditions, loss of *Atg7* in this neuronal population has no effect on body weight, food intake, or energy expenditure. However, when fasted overnight, mice lacking *Atg7* displayed reduced O_2_ consumption, CO_2_ and heat production, and reduced food intake after refeeding. *Atg7* deletion in SF1 neurons only implied moderate changes in fed plasma insulin levels and insulin resistance, without changes in glycemia and glucose tolerance.

This study demonstrates autophagy activation in the VMH, in addition to other regions such as the ARC, in response to fasting. This was supported not only by increased expression of autophagy-related proteins in the area but also by a 2- to 3-fold increase in the density of cFos-immunoreactive cells in the VMH, ARC, and DMH of control mice in response to fasting, an induction that was abrogated in these three regions in *Atg7* deficient mice in SF1 VMH neurons [61]. Then, the lack of autophagy in SF1 VMH neurons significantly impacts neurons outside of the VMH. This is in line with previous investigations showing that POMC neurons in the ARC receive excitatory inputs from the VMH, and the strength of this VMH-ARC input is reduced by fasting [84].

Loss of autophagy in SF1 VMH neurons also caused alterations in mitochondrial function and morphology, which was associated with inadequate metabolic response to fasting [61]. During fasting, mitochondrial efficiency must increase, and autophagy is important to ensure the degradation of mitochondria that are damaged beyond repair, that is, the process called mitophagy [85]. Mice lacking *Atg7* in SF1 neurons showed a reduced number of mitochondria-autophagosome contacts with alterations in mitochondria morphology and respiration, indicating abnormal mitophagy [61]. Since UCP2 expression in SF1 VMH neurons has been linked to mitophagy [62,86], the authors explored the expression of this protein, and they found it was dysregulated in mutant mice [61]. In agreement with these findings, UCP2 in SF1 VMH regulates mitochondria fission and the excitability of GE neurons in the VMH, which in turn allows the appropriate response to increased glycemia, enhancing insulin sensitivity in peripheral tissues [62]. However, the moderate effects of autophagy loss in SF1 neurons observed in glucose homeostasis do not correspond to the abnormal expression of UCP2 detected. The fact that UCP2-induced mitochondrial fission in the VMH only affects GE neurons, but not GI neurons [62], could explain the moderate phenotype of the mice deficient in *Atg7* in glucose metabolism.

Very recently, the SF1 neurons’ primary cilia, a solitary antenna-like extension of the plasma membrane, was demonstrated as an important organelle for the regulation of energy homeostasis [63]. In this study, the authors deleted the intraflagellar transport 88 (IFT88), which is a critical protein in primary cilium biogenesis, specifically in SF1 neurons. Dysfunction of VMH primary cilia resulted in impaired activation of sympathetic tone, central leptin resistance, and higher body weight gain under both chow and HFD feeding. Obesity in the transgenic mice IFT88-KO^SF−1^ was caused by a marked decrease in energy expenditure, hyperphagia (only under HFD feeding), blunted BAT function, insulin resistance, and glucose intolerance. In addition, deletion of primary cilia in SF1 neurons led to an impairment in bone homeostasis. The downregulation of the sympathetic activity in IFT88-KO^SF−1^ mice was linked to the reduction of the metabolic rate and increase in bone density, as this effect considered was independent of the obese phenotype [63]. Considering the fact that primary cilia are required for proper neuronal circuit formation, deletion of this organelle in SF1 neurons may therefore influence local neuronal circuits [63].

## 4. The Sex-Specific Effect of SF1 Neurons on Energy Balance

It is known that the VMH is sexually dimorphic, showing females higher ERα concentration than males [87]. Additionally, as described in Section 3, selective deletion of this receptor in SF1 neurons resulted in increased abdominal obese phenotype with adipocyte hypertrophy in females, but not in males [52]. Obesity in females was caused by decreased energy expenditure as they had reduced basal metabolic rate and impaired BAT thermogenesis [52]. It has been described also that estrogens regulate the activity of GI neurons of the ventrolateral portion of VMH, since females showed an attenuated response to hypoglycemia compared to male mice, although in this study it is not specified if these neurons were SF1 positive cells [88]. As has been already specified, there are other genetic deletions that presented different outputs in a sex-specific manner. Expression of mGluR5 in SF1 neurons was necessary for estradiol protective effects in glucose balance in female but not in male mice, and mGluR5 deletion resulted in reduced electrical activity only in female mice [54]. Deletion of α2δ-1 expression in SF1 neurons also presented sexually dimorphic effects depending on diet conditions. Particularly, female mice lacking α2δ-1 displayed glucose intolerance and insulin resistance under chow or HFD, as this phenotype much moderate in male mice [58,59]. Cheung et al. also observed sex-dependent changes when deleting VGLUT2 in SF1 neurons, since female but not male mice presented attenuation of DIO, and transgenic male mice showed behavioral changes not observed in female mice [89].

Estrogens could affect and change some intracellular signaling cascades leading to these differences observed between males and females, as female mice have greater expression of ERα. This would explain why the presence or absence of different patterns of receptors impacts the estrogenic effect on SF1 neurons, as some of the sex-specific signaling cascades are being altered. Other sex-specific effects related to SF1 neurons can be found in their synapses with POMC neurons, as estradiol attenuated the retrograde endocannabinoid signaling from POMC to SF1 neurons, increasing the glutamatergic inputs to POMC [78]. This study opens another possibility based on the fact hypothesis that estrogens do not interact directly with the specific SF1 neurons targeted in each study, but their effects fall on other cells acting on SF1 neurons.

Then, estrogen signaling in SF1 neurons is a must for metabolic health in female mice. Despite the fact that many studies presented in this review were performed only in male mice, these last results obtained in both sexes reinforce the idea of SF1 neurons having a sex-specific effect on energy balance and glucose metabolism.

## 5. Exploring the Neurocircuitry That Links SF1 Neurons to Other Brain Areas in Energy Balance

Remarkable progress in the neuroanatomy of VMH projections has been obtained from novel biological tools. Some notions on how the VMH connects with other brain centers come from stereotaxic injection with anterograde axonal tracers in adult rats [90,91]. While the results from these studies established that VMH is organized in subregions depending on their projection, the inherent limitations of this method make it difficult to assess the specific neuronal network involved in SF1 neurons. To overcome these limitations, *Sf-1^TauGFP^* and *Z/EG^Sf1:Cre^* mice models were originally designed to trace the major VMH axonal projections during embryonic and postnatal stages [9].

The first of these models is a knock-in line that contains the wheat germ agglutinin (WGA) and Tau-green fluorescent protein (TauGFP) under the control of *Sf-1* regulatory elements. In the second model, the SF1 Cre mouse was crossed with a *Z/EG* reporter mouse, resulting in constitutive expression of eGFP (enhanced GFP) after Cre-mediated recombination [9]. The analysis of the results obtained from both models indicated the efferent SF1 projection in ascending and descending tracts that innervate different structures such as the hypothalamus, thalamus, the basal forebrain, and the brainstem. Interestingly, SF1 projections targeting the general vicinity of gonadotropin-releasing hormone (GnRH) neurons suggest the potential role in fertility physiology [9].

As expected, experiments using these models showed that SF1 neurons projected to areas implicated in body weight regulation, including the paraventricular nucleus of the hypothalamus (PVN) [9,92]. However, it was not possible to detect any GFP^+^ fibers within the ARC. Thus, additional studies using synaptophysin or rabies virus would be helpful to explore the specific network between SF1 and hypothalamic nuclei involved in energy balance. Recently, Yunglei Yang and colleagues have optogenetically identified the downstream target underlying the SF1 suppression of food intake (Figure 5). The authors used the SF1 ChR2 mice, but the fiber optic cannula was implanted above the PVN, therefore the photostimulation of this area would only activate the SF1-PVN projections. They showed that high-frequency stimulation of this circuit potently reduced food intake even in 24 h food-deprived mice [23].

Downstream projections of SF1 neurons regulating glycemia were identified through the same novel strategy derived from optogenetics. The researchers administrated fluorescently tagged ChRs into the VMH of SF1-Cre mice to visualize the projection fields using histological imaging to detect the EYFP reporter in axonal projections [32]. Once identified this target site, a fiber optic was implanted to depolarize the final projection by laser stimulation. With this strategy, it was possible to demonstrate that VMH^SF1^→aBNST is the most relevant circuit involved in controlling glycemia since the photostimulation of this circuit triggered an increase of the blood glucose levels [32] (Figure 5). Interestingly, other functions have been attributed to this specific circuit: Fan Yang and colleagues recently demonstrated that SF1 neurons are innervated by upstream BNST neurons and send projections to the NST nucleus to regulate anxiety-like behavior and bone metabolism [93] (Figure 5).

The use of current and novel biological tools provides several clues about the neurocircuits regulating blood glucose and energy balance, and the afferent and efferent connections linking SF1 neurons with other neuronal populations. An improving understanding of the functional organization of SF1 neurons may help to identify future strategies for metabolic diseases such as obesity and diabetes.

## 6. Concluding Remarks and Future Perspectives

In the last years, the critical role played by the hypothalamus in the regulation of energy balance and glucose homeostasis has gained substantial importance. However, the exact mechanisms and neuronal circuits underlying this regulation and the pathogenesis of obesity and insulin resistance remain poorly understood. The growing literature is demonstrating that the origin of these diseases is beyond feeding and pancreatic insulin secretion, and needs further definition. Here we extensively review that SF1 neurons in the hypothalamus provide a “central role” in the control of blood glucose levels, insulin sensitivity in peripheral tissues, adipose tissue plasticity, and thermogenesis activation. SF1 neurons are a specific lead in the brain’s ability to sense glucose levels and conduct insulin and leptin signaling in energy expenditure and glucose homeostasis, with minor feeding control. Interestingly, transgenic mice lacking different targets in SF1 neurons show an altered metabolic phenotype mostly under HFD feeding, but not under chow diet, indicating that SF1 neurons are particularly important for metabolic adaptation in the early stages of obesity. These investigations also demonstrate the sex-specific effects of SF1 neurons in the VMH, and the importance of the analysis of the phenotype in both male and female mice when exploring energy balance and metabolism in transgenic models.

Although optogenetic and chemogenetic tools have clearly demonstrated SF1 function in glucose homeostasis, adiposity, and energy expenditure, there are still contradictory results in these functions after deletion of specific receptors in SF1 neurons (e.g., mGluR5). These controversies could be associated with the existence of several sub-populations of SF1 neurons that respond differently to insulin, leptin, and glucose. For instance, whereas some SF1 neurons seem to be specialized in the regulation of blood glucose levels, some others are responsible for insulin sensitivity in the periphery.

When investigating the hypothalamic regulation of obesity and diabetes, in addition to the identification of SF1 sub-populations, it is important to explore the coordination of SF1 neurons with other neurons outside the VMH to trigger metabolic functions. For instance, HFD-induced hyperinsulinemia drives SF1 neurons hyperpolarization, leading in turn to functional changes in the synaptic output onto POMC neurons, causing obesity and glucose intolerance. This evidence suggests that the nucleus-specific responses upon HFD feeding can cooperate to cause obesity and diabetes, and disruption of this cooperation could act as a link between both diseases.

Despite the advances in the neurocircuitry connecting SF1 VMH neurons with other brain areas, the exact efferent circuits leading to changes in peripheral tissues need further investigation. How mice lacking a protein only in SF1 neurons present a strong phenotype in the periphery in terms of glucose homeostasis and energy expenditure under nutritional challenges? Are these circuits restricted to SNS and PSNS? A new point of view also emerges from these studies: a disruption in a specific type of neurons in the hypothalamus could trigger the pathogenesis of type 2 diabetes and obesity. How this neuronal dysfunction drives peripheral insulin resistance, adiposity, and thermogenesis alteration remains unclear. Reconsidering the pathogenesis of obesity and type 2 diabetes and now including the essential role of SF1 neurons, new strategies to treat these diseases could emerge. Targeting this specific population of neurons in the VMH, and even another cooperating population outside the VMH at the same time could modulate specific metabolic functions, a challenging approach but with significant translational impacts in the management of obesity and diabetes.

## Figures and Tables

**Figure 1 ijms-22-06186-f001:**
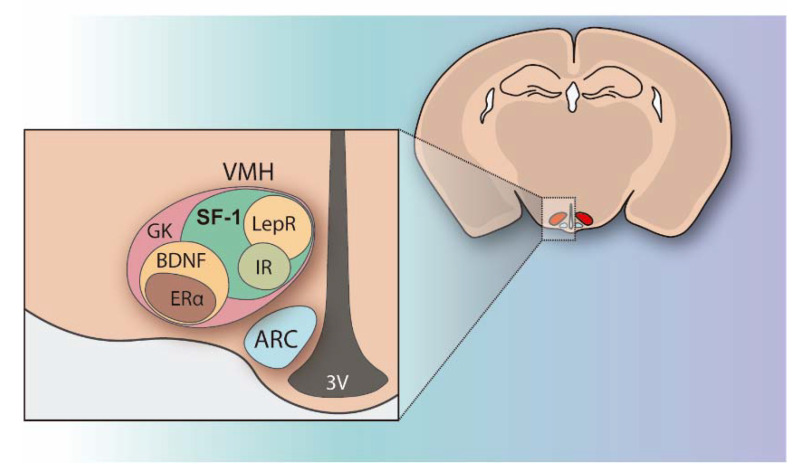
Schematic illustration of the pattern of genes highly expressed in the VMH. The majority of VMH cells, especially in the dorsomedial and central regions of VMH, express the nuclear receptor steroidogenic factor 1 (SF1). Leptin receptor (LEPR) positive cells mainly converge in the dorsomedial part, whereas insulin receptor (IR) maps the central region. Cells expressing brain-derived neurotrophic factor (BDNF) are mainly distributed in central and lateral areas of the VMH, estrogen receptor (ERα)-expressing cells are limited to the lateral region, and glucokinase (GK)-positive cells are present throughout the VMH. This schematic diagram is based on previous articles from Choi et al. [8] and Yi et al. [6].

**Figure 2 ijms-22-06186-f002:**
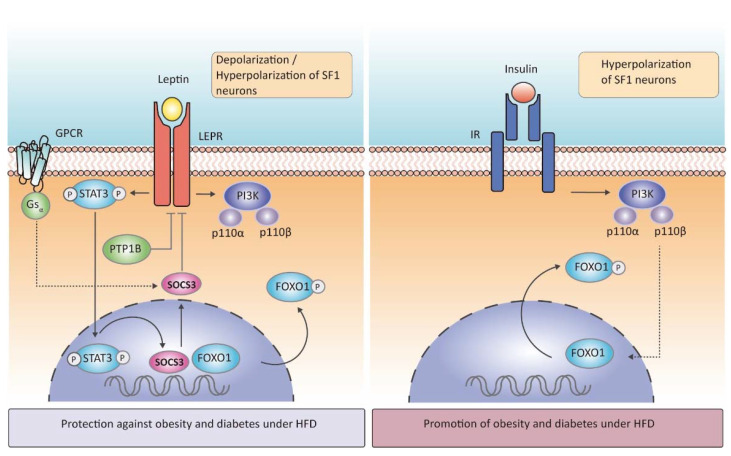
Leptin and insulin signaling in SF1 neurons. Leptin binds LEPR to regulate body weight gain, energy expenditure, and insulin sensitivity in the VMH via STAT3 phosphorylation which is translocated into the nucleus and induces the expression of genes such as SOCS3, a negative regulator of LEPR. Leptin also stimulates a PI3K-dependent pathway, particularly involving subunits p110α, which contributes to energy expenditure regulation, and p110β, which contributes to both brown fat thermogenesis and glucose metabolism regulation in SF1 neurons. The Gsα subunit of G protein-coupled receptors (GPCR) of hormones and the protein-tyrosine phosphatase 1B (PTP1B) negatively regulates leptin action. The activation of this leptin signaling pathway triggers protection against obesity and diabetes under a high-fat diet (HFD) feeding. According to insulin signaling in SF1 neurons, exposure to HFD drives hyperinsulinemia, insulin binds insulin receptor (IR), which in turn activates PI3K that indirectly phosphorylates FOXO1 to induce transcription factors and promote obesity and glucose metabolism disruption.

**Figure 3 ijms-22-06186-f003:**
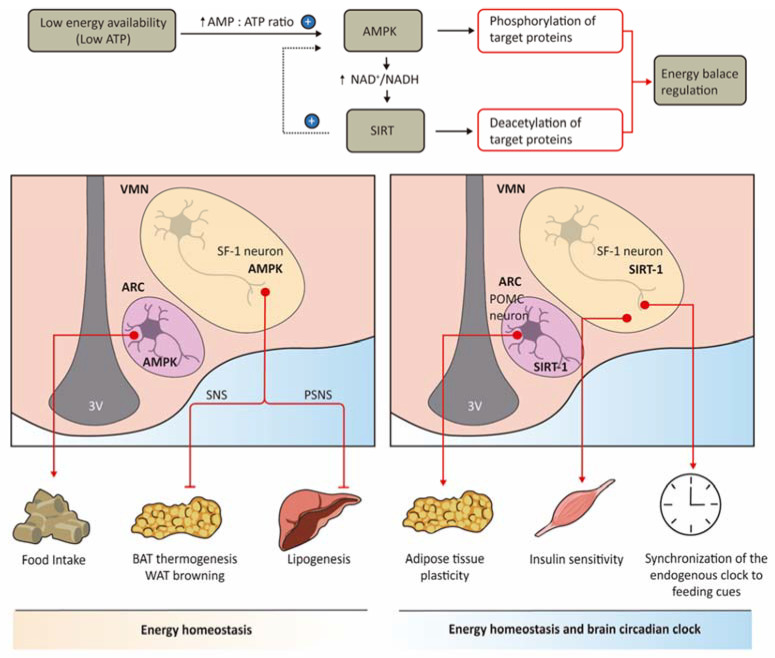
Nutrient sensors in SF1 neurons regulate energy homeostasis. AMPK and SIRT1 are key nutrient sensors that restore energy balance under low energy availability. The role of both AMPK and SIRT1 in the central regulation of energy balance depends on their specific location in hypothalamic nuclei. AMPK in SF1 neurons of the VMH particularly regulates energy expenditure by acting in BAT thermogenesis and subcutaneous WAT browning (via sympathetic output) and liver lipogenesis (via parasympathetic output), whereas AMPK effects in feeding control and glucose balance mostly rely on ARC neurons. SIRT1 in SF1 VMH neurons controls insulin sensitivity in skeletal muscle protecting against diet-induced obesity, but in POMC neurons of the ARC, SIRT1 regulates normal BAT-like remodeling in specific fat depots. SIRT1 in SF1 neurons also contributes to the synchronization of the circadian clock in the brain by nutritional inputs. The arrow means increase and + means activation.

**Figure 4 ijms-22-06186-f004:**
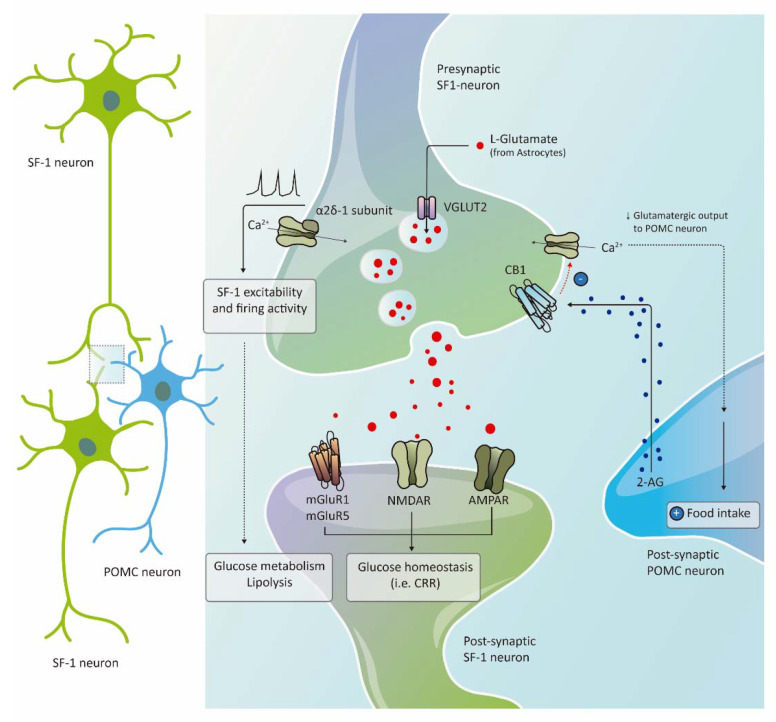
Glutamatergic neurotransmission and synaptic receptors in SF1 neurons and energy balance. Lack of the glutamate vesicular transporter VGLUT2 in SF1 neurons disrupts the glutamate effect in the postsynaptic receptors, leading to glucose metabolism dysfunction and a defective counterregulatory response (CRR). In line with this, the glutamate receptor mGluR5 in SF1 neurons also regulates glucose balance. The calcium channel subunit α2δ-1 is essential for excitability and firing activity of SF1 neurons which in turn regulate glucose metabolism in adipose tissue and skeletal muscle and lipolysis in white adipose tissue with no alterations in food intake. Finally, the endocannabinoid 2-arachidonoylglycerol (2-AG) released in the post-synaptic neuron, leads to retrograde activation of the CB1 receptor located in the presynaptic glutamatergic SF1 neurons, blocking calcium entry with subsequent inhibition of glutamatergic output and attenuation of POMC neurons activation, promoting food intake in diet-induced obesity.

**Figure 5 ijms-22-06186-f005:**
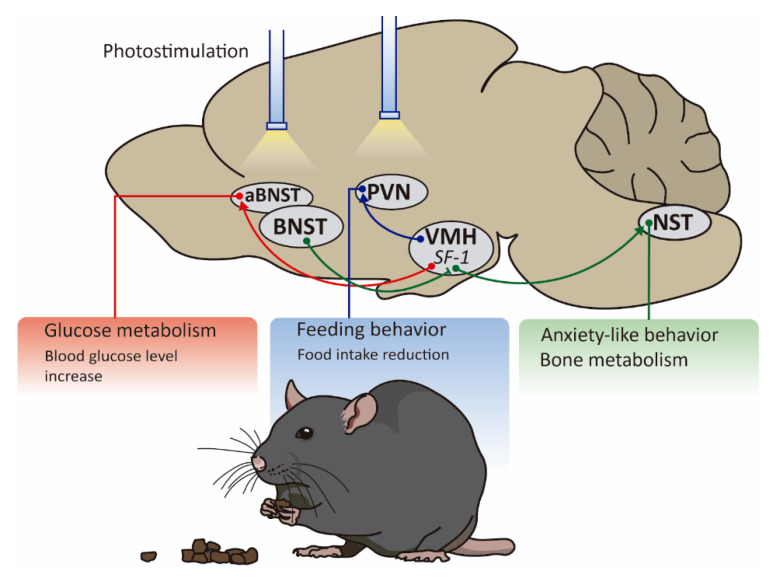
Neurocircuitry linking SF1 VMH neurons to other brain areas in energy balance. Using optogenetic tools, the paraventricular nucleus of the hypothalamus (PVN) has been identified as a downstream target nucleus underlying the SF1 suppression of food intake. Downstream projections of SF1 in the VMH involved in the control of glycemia are the bed nucleus of the stria terminalis (aBNST) since photostimulation of the VMH^Sf1^-aBNST circuit leads to increases in blood glucose levels. Finally, SF1 neurons are innervated by upstream BNST neurons and send projection to the nucleus tractus solitarius (NST) to regulate anxiety-like behavior and bone metabolism.

## Data Availability

Not applicable.

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
