# Peer review of "New Insights of SF1 Neurons in Hypothalamic Regulation of Obesity and Diabetes"

_ijms, 2021, doi:10.3390/ijms22126186_

Round 1
Reviewer 1 Report
Fosch and Collegues collect in this review the more recent and relevant studies about SF1 neurons and their role in energy metabolism and correlated pathologies. The authors seems to start when the previous review of Choi et al finish (2013) adding important new data on the new transgenic lines and optogenetic / chemogenetic approaches.
In my opinion the review is well written and interesting, and the figures are nice.
I have only some suggestions and a minimum criticism for improving the manuscript.
I think that the understanding of the topic would be improved if the authors add an initial figure of a coronal section of the hypothalamus with the VMH (something similar to that made by CHOI et al, 2013 will be perfect), with the subdivision of the neuronal populations.
In row 36-37
…Therefore, understanding the molecular and physiological mechanisms underlying the control of feeding behavior, energy balance and glucose homeostasis is key for the prevention and treatment of obesity and diabetes”
Use is a key for or replace the word key with “fundamental” or something similar.
In TABLE 1 please consider that if the table will be split on two pages it would be better to repeat the header / caption to facilitate reading.
Moreover, in the table it is very interesting that it is highlighted whether the studies are done on males or females or on both sexes, but in the text there is only few references to the fact that the studies were done only in one sex (i.e. starting from row 278); considering that SF1 neurons may have a sex-specific effect on energy balance and glucose metabolism perhaps the authors could better comment this lost of information (in a separate paragraph or maybe in the concluding remarks?) underlining the importance of an approach based on sex differences considering the problem of food intake and energy balance.
In row 333
SF1-Cre AMPKα1flox/flox mice fed a long-term HFD had a feeding-independent decrease in body weight and adiposity…
Add with before a long-term HFD…
The same in row 348: fed with a HFD; and in row 434
Author Response
Fosch and Collegues collect in this review the more recent and relevant studies about SF1 neurons and their role in energy metabolism and correlated pathologies. The authors seems to start when the previous review of Choi et al finish (2013) adding important new data on the new transgenic lines and optogenetic / chemogenetic approaches.
In my opinion the review is well written and interesting, and the figures are nice.
I have only some suggestions and a minimum criticism for improving the manuscript.
We would like to thank the Reviewer for the time taken to prepare the comments. On the whole we have found the comments very constructive and have tried to address each suggestion in turn. Page numbers are referred to the new version of the manuscript and all the changes are highlighted in red.
- I think that the understanding of the topic would be improved if the authors add an initial figure of a coronal section of the hypothalamus with the VMH (something similar to that made by CHOI et al, 2013 will be perfect), with the subdivision of the neuronal populations.
In agreement with the Reviewer, this initial figure can clarify the content of the review. Then, a schematic figure has been now included (New Figure 1), based on the studies from Choi et al (2013) and Yi et al (2006).
- In row 36-37 “…Therefore, understanding the molecular and physiological mechanisms underlying the control of feeding behavior, energy balance and glucose homeostasis is key for the prevention and treatment of obesity and diabetes”. Use is a key for or replace the word key with “fundamental” or something similar.
We have changed this by “crucial” (row 38-39).
- In TABLE 1 please consider that if the table will be split on two pages it would be better to repeat the header / caption to facilitate reading.
As suggested by the Reviewer, we have repeated the header on the second page of Table 1.
- Moreover, in the table it is very interesting that it is highlighted whether the studies are done on males or females or on both sexes, but in the text there is only few references to the fact that the studies were done only in one sex (i.e. starting from row 278); considering that SF1 neurons may have a sex-specific effect on energy balance and glucose metabolism perhaps the authors could better comment this lost of information (in a separate paragraph or maybe in the concluding remarks?) underlining the importance of an approach based on sex differences considering the problem of food intake and energy balance.
We thank the Reviewer for this important comment. We also consider the importance of the sex-dependent effects of SF1 neurons on energy balance. Then, we have incorporated a specific section for this issue (New Section 4, rows 603-636), and also a sentence in the Conclusion section (rows 709-712).
- In row 333…”SF1-Cre AMPKα1flox/flox mice fed a long-term HFDhad a feeding-independent decrease in body weight and adiposity…” Add with before a long-term HFD… The same in row 348: fed with a HFD; and in row 434
These changes have been done.
Reviewer 2 Report
This review articles include valuable scientific contents especially for SF-1 neurons function in the regulation of energy homeostasis. The manuscript is well-focused and –written, so I have only one suggestion.
Minor suggestions:
- There is a recent publication elegantly addressed VMH neurons and cellular organelle in the regulation of feeding and energy homeostasis published in JCI (J Clin Invest. 2021 Jan 4;131(1):e138107. doi: 10.1172/JCI138107). The authors may want to extend their information as they put this reference in “Table 1”
Author Response
This review articles include valuable scientific contents especially for SF-1 neurons function in the regulation of energy homeostasis. The manuscript is well-focused and –written, so I have only one suggestion.
Minor suggestions:
- There is a recent publication elegantly addressed VMH neurons and cellular organelle in the regulation of feeding and energy homeostasis published in JCI (J Clin Invest. 2021 Jan 4;131(1):e138107. doi: 10.1172/JCI138107). The authors may want to extend their information as they put this reference in “Table 1”
We thank the Reviewer for the positive comments on our manuscript. As suggested by the Reviewer, we have now incorporated this interesting and recent study published by Sun et al. (JCI 2021) (see Table 1, indicated in red lettering, and Section 3.4 (rows 586-600).